# Comprehensive Transcriptome Analysis Uncovers Distinct Expression Patterns Associated with Early Salinity Stress in Annual Ryegrass (*Lolium Multiflorum* L.)

**DOI:** 10.3390/ijms23063279

**Published:** 2022-03-18

**Authors:** Guangyan Feng, Pengqing Xiao, Xia Wang, Linkai Huang, Gang Nie, Zhou Li, Yan Peng, Dandan Li, Xinquan Zhang

**Affiliations:** Department of Forage Science, College of Grassland Science and Technology, Sichuan Agricultural University, Chengdu 611130, China; feng0201@sicau.edu.cn (G.F.); xpq1227811090@163.com (P.X.); wangxiayuyao@126.com (X.W.); huanglinkai@sicau.edu.cn (L.H.); nieg17@sicau.edu.cn (G.N.); lizhou1986814@163.com (Z.L.); pengyanlee@163.com (Y.P.); lidandan@sicau.edu.cn (D.L.)

**Keywords:** early salinity stress, annual ryegrass, TFs, phytohormones, salt stress sensing

## Abstract

Soil salination is likely to reduce crop production worldwide. Annual ryegrass (*Lolium multiflorum* L.) is one of the most important forages cultivated in temperate and subtropical regions. We performed a time-course comparative transcriptome for salinity-sensitive (SS) and salinity-insensitive (SI) genotypes of the annual ryegrass at six intervals post-stress to describe the transcriptional changes and identify the core genes involved in the early responses to salt stress. Our study generated 215.18 Gb of clean data and identified 7642 DEGs in six pairwise comparisons between the SS and SI genotypes of annual ryegrass. Function enrichment of the DEGs indicated that the differences in lipid, vitamins, and carbohydrate metabolism are responsible for variation in salt tolerance of the SS and SI genotypes. Stage-specific profiles revealed novel regulation mechanisms in salinity stress sensing, phytohormones signaling transduction, and transcriptional regulation of the early salinity responses. High-affinity K^+^ (HAKs) and high-affinity K1 transporter (HKT1) play different roles in the ionic homeostasis of the two genotypes. Moreover, our results also revealed that transcription factors (TFs), such as WRKYs, ERFs, and MYBs, may have different functions during the early signaling sensing of salt stress, such as WRKYs, ERFs, and MYBs. Generally, our study provides insights into the mechanisms of the early salinity response in the annual ryegrass and accelerates the breeding of salt-tolerant forage.

## 1. Introduction

Soil salinization is a dire threat to sustainable agricultural production and ecological security globally [1]. Consequently, excess salt significantly reduces grain yield and biomass production of most crop plants [2]. Soil salinization has been accelerating because of climate change and irrational irrigation and land utilization. In China, over 99.13 million hectares (~10% land area), including 90% of the inland and about 8.5% of the coastal soil, have been affected by salinization [3]. Currently, research on saline stress response has been performed in rice (*Oryza sativa* L.) [4], wheat (*Triticum aestivum* L.) [5], maize (*Zea mays* L.) [6], soybean (*Glycine max* L.) [7], and barley (*Hordeum vulgare* L.) [8]. Nevertheless, studies on salt tolerance in forage crops are limited. Therefore, understanding the complex salinity resistance mechanisms is important for selecting suitable salt-tolerant crops for the efficient utilization of salt-affected soils.

Soil salinization may lead to several stresses on crops, such as K^+^/Na^+^ imbalance, osmotic stress, nutrient imbalance, and oxidative stress [9]. Moreover, excessive uptake of Na^+^/Cl^-^ and loss of K^+^ results in physiological drought and ionic toxicity, interfering with enzymes stability and protein synthesis in plants [10]. Generation and scavenging of reactive oxygen species (ROS) are also hampered under saline conditions. Overaccumulation of ROS damages proteins and the cellular phosphate layer, inducing senescence and death of plant cell [11]. The ROS antioxidant system is mainly composed of superoxide dismutase (SOD), peroxidase (POD), and catalase (CAT), which offset the oxidative damages to a certain degree [12]. Both salt-sensitive and salt-tolerant plants have developed adaptive response mechanisms to cope with the adverse effects of soil salinization. Various signaling pathways are individually or synergistically involved in these stress response processes. Plants sense salt stress through ionic and osmotic signals, which are transferred to the cell body through plasma membrane receptors. Previous studies have demonstrated that ROS also act as a secondary messenger, trigging plant adaptive responses to high salt concentration in soil [13,14]. Salinity stress leads to ROS accumulation and the synthesis of ABA [15]. Moreover, plant hormones, including ABA, auxin, cytokinins, and ethylene, function in signaling transition and root system architecture, which enhance salt stress tolerance [10,16]. Analogously, Ca^2+^ affects plant growth via the Ca^2+^-induced ion channel discrimination against Na^+^ [17]. The CBL-interacting serine/threonine-protein kinase 1 (CIPK) and calcium-binding protein (CML) have been shown to mediate a crosstalk between calcium ion signaling and salt overly sensitive (SOS) signaling [18]. The SOS pathway is affected by the increasing ABA synthesis induced by the ABI2 protein. It has been reported that diverse signals directly or indirectly converge at MAPK cascades and that key genes, such as *MAPKKKs*, *CDPKs*, *HK1*, *LEA-TYPE*, *bZIP*, and *MYB*/*MYC*-type genes, offer potential nodes for the gene regulatory networks [19] These genes activate complex molecular responses at the transcriptional and translation levels to maintain cellular environmental homeostasis [9,20].

Annual ryegrass (*Lolium multiflorum* L.) is one of the most important forages cultivated in temperate and subtropical regions. It is utilized as pasture or hay grass due to its high biomass yields, excellent palatability, and high adaptability. The potential of the annual ryegrass as pioneer plant for forage production in saline land provides a new insight into the use of marginal land. Previous studies have demonstrated that higher concentrations of ions, especially NaCl, in saline soil cause the abnormal growth of plants [21]. In our previous study, two annual ryegrass varieties (Diamond T× Yangtze River 2 (Salinity-insensitive genotype, SI) and Tetragold (Salinity-sensitive genotype, SS)) with different degrees of salt tolerance were developed [22]. However, a few studies have examined the influence of salt stress in the annual ryegrass at the molecular level. This study aimed at conducting a time-course comparative transcriptome profiling to describe the transcriptional changes and identifying the core genes involved in the early stages of salt stress responses in the annual ryegrass. The results will provide insights into the regulatory mechanisms underlying salt stress response in annual ryegrass and accelerate breeding of salt-tolerant varieties for forage utilization of the marginal land.

## 2. Results

### 2.1. Phenotypic and Physiological Responses of L. Multiflorumi to Salt Stress

We subjected the leaf samples to 300 mM NaCl treatment and collected data at different time points (0, 0.5, 2, 6, 12, 24, 48, and 72 h). There was an increase in SOD and POD activities after 0.5 h, which slightly decreased after 0.5 h to 6 h, followed by a sharp increase after 12 h to 72 h. Similarly, CAT activity increased sharply from 0 h to 2 h, then declined after 6 h to 12 h, and increased again after 12 h to 72 h (Figure 1A–C). Relative water content (RWC) declined gradually after increasing between 0.5 h and 2 h (Figure 1E). The SOD, POD, and CAT activities exhibited a significant difference between salinity-insensitive and salinity sensitive accessions at the last two sampling points (48 h and 72 h); however, the salinity-insensitive accession values were much higher than the salinity sensitive accession ones. The change trends of MDA content and electrolyte leakage were consistent with those of the SOD, POD, and CAT activities at the different sampling points (Figure 1D,F). However, MDA content and electrolyte leakage were significantly higher in salinity-sensitive accession than in salinity-insensitive accession after 24 h and 48 h of salt treatment. Furthermore, the salinity-insensitive accession group exhibited a lower Na^+^/K^+^ ratio after 12 h salt treatment (Appendix A), suggesting stronger salt tolerance. Generally, the patterns of these three physiological indicators (SOD, POD, and CAT) showed two peaks, indicating a prompt response by ryegrass to salt stress, which increases upon adapting to the adverse conditions (Figure 1A-C).

### 2.2. Processing and Mapping of the RNA-Seq Data

Thirty-six cDNA libraries were prepared using samples (triplicate) obtained from the salinity-sensitive and -insensitive genotypes after 0, 0.5, 2, 6, 12, and 24 h of the treatment to determine the transcriptional dynamic of exposing *L. multiflorum* to short-term saline stress. Principal component analysis revealed that 36 samples clustered into six discrete groups and that the samples clustered between 0 h and 0.5 h showed a shorter distance than others (Figure 2A). After normalization, 1,434,659,786 clean reads with 215.18 Gb of clean bases were generated. The total clean data for each sample ranged from 36.1 to 42.7 million kilobases (Appendix A). The clean data were assembled into 103,680 unigenes (>200 bp) with an N50 of 1522 bp. The minimum length of the unigenes was 201 bp, and maximum length of unigenes was 15,032 bp (Appendix A). Over 80% of the assembled unigenes were mapped, resulting in 75% of the uniquely mapped unigenes and 5.19% to 7.54% of the multiple mapped unigenes (Appendix A). In total, 52,086 unigenes were annotated in at least one of the databases (Nr, KEGG, KOG and SwissProt), of which 19,344 unigenes were annotated in all databases (Appendix A). The FPKM values of all the 36 samples were obtained via violin plot (Figure 2B), and our data were confirmed to be reliable for the subsequent analyses. All the sequenced raw data were submitted to the NCBI database using accession number SRR16085492, SRR16085494, SRR16085487, SRR16085488, SRR16085493, SRR16085489, SRR16085497, SRR16085496, SRR16085491, SRR16085490, SRR16085498, and SRR16085495.

### 2.3. Identification of the Differentially Expressed Genes in the SS and SI Genotypes of L. Multiflorum under Salt Stress

To understand the transcriptional difference between the control and treatment groups after 0.5, 2, 6, 12, and 24 h of treatment, we identified 7642 DEGs in six pairwise comparisons. These included; 1) 965 genes (584 up-regulated and 381 down-regulated) in SS0 versus SI0, 2) 1077 genes (436 up-regulated and 641 down-regulated) in SS0.5 versus SI0.5, 3) 1380 genes (696 up-regulated and 684 down-regulated) in SS2 versus SI2, 4) 1359 genes (537 up-regulated and 822 down-regulated) in SS6 versus SI6, 5) 1587 genes (707 up-regulated and 880 down-regulated) in SS12 versus SI12, and 6) 1274 genes (519 up-regulated and 755 down-regulated) SS24 versus SI24 (Figure 3A). The DEGs increased from 0 h to 12 h, then declined after 24 h following the salt treatment. The highest number of DEGs and down-regulated DEGs were identified 12 h post-stress treatment, implying that the transcriptional expression pattern changed rapidly at the early phases of salinity stress. More genes were down-regulated than up-regulated in the SS genotype compared to the SI genotype, especially at 0.5, 6, and 24 h post-salt stress. However, 33 genes were differentially expressed across the two genotypes, with the expression exhibiting either continuous up-regulation or down-regulation patterns in the SI (19 DEGs) and SS (14 DEGs) genotypes (Figure 3C). Several genes associated with plant stress responses were identified among these DEGs, including *MYB73, ARF1, RGA5,* and *RPPL1* in the SI genotype, and *MADS51* in the SS genotype. Additionally, the highest number of DEGs was identified between 12 h and 24 h, while the lowest number was detected between 6 h and 12 h in both SS and SI genotypes after the treatment (Figure 3B). The DEGs increased tardily, then declined sharply between 6 h and 12 h, and later peaked between 12 h and 24 h in both SS and SI genotypes (Figure 3D). The changing trend of the number of DEG was consistent with the changes in several physiological indexes, hinting at a possible association between transcriptional and physiological reactions.

Genes linked to Na^+^/K^+^ were also detected in our RNA-seq data. They included: (1) five sodium/hydrogen exchangers (*NHX*s), (2) one cation transporter (*HKT1*), (3) one cations/H^+^ antiporter (*CHX20*), (4) 10 potassium channel-related genes, (5) 16 potassium transporters (*HAKs*), (6) three vacuolar cation/proton exchangers (*CAXs*), (7) five chloride channel proteins (*CLCs*), and (8) 12 cyclic nucleotide-gated ion channel-related genes (*CNGCs*) (Appendix A). At 2 h post salinity stress, only *HKT1* was differentially expressed between the SS and SI genotypes. Additionally, the expression of most *NHXs* was higher in the SS genotype compared to the SI, but no significant differences were observed among the different sampling points.

### 2.4. Functional Characterization of the DEGs

The GO and KEGG annotations were performed to understand the functions of the DEGs obtained from the SS and SI genotypes. The DEGs were significantly enriched in several GO categories, which have been associated with plants salinity tolerance. These included cell periphery; cell/cell–cell junction; and responses to oxidative stress, metal ions, reactive oxygen species, and response to inorganic substances. In contrast, the GO annotations of the DEGs had great variations at different sampling points. After 0.5 h of the salt treatment, the cell periphery, external encapsulating structure, and anatomical structure development were the top three enriched GO categories (Appendix A). Meanwhile, the respiratory electron transport chain, electron carrier activity, and oxidation–reduction process were significantly enriched after 2 h of the salt treatment (Appendix A). Significant enrichment of the responses to inorganic substance, metal ion, and oxidative stress was obtained between 6 h and 12 h (Appendix A). Finally, the ribosome, ribonucleoprotein complex, and non-membrane-bounded organelle were enriched after 24 h of the salt treatment (Appendix A). Generally, these results revealed the genes involved in cellular components and physiological responses during the early stages of salinity stress. Environmental stimulus signals and the biological transport processes were also induced by extending the salinity treatment duration. Additionally, the annual ryegrass maintained a certain degree of growth under salinity stress following a complex molecular regulation and physiological coordination between its intracellular and extracellular environment.

The DEGs obtained at 0.5 h, 2, 6, 12, and 24 h were assigned to KEGG pathways, including 90 pathways at 12 and 24 h each, 89 pathways at 6 h, 83 pathways at 0.5 h, and 74 pathways at 2 h. However, there were differences in the number of significantly enriched pathways. Many significantly enriched pathways were identified at 12 h (13 pathways) and 0.5 h (8 pathways) following the treatment, relative to 2 h (3 pathways), 6 h (3 pathways), and 24 h (5 pathways). Unlike the GO annotation, most sampling points had similar top-five terms in the KEGG categories. Ribosome and oxidative phosphorylation categories were identified as the top three pathways at 6, 12, and 24 h (Appendix A), while the metabolic pathways were significantly enriched at 0.5 and 2 h (Appendix A). This revealed the importance of translation and energy metabolism in salt stress response. Moreover, the enrichment of the transcription and replication, environmental adaptation, and signal transduction pathways after the salinity treatment showed the differences in salt response between SS and SI genotypes.

### 2.5. Weighted Gene Co-Expression Network Analysis of the DEGs

Weighted gene co-expression network analysis (WGCNA) was conducted to determine the main genes implicated in the early responses of different annual ryegrass genotypes. After normalizing the microarray expression data, 17,180 probes were selected from 35 samples (based on the pairwise correlations) for co-expression networks construction. The specific modules containing a series of genes with high correlations were presented using distinct colors. Fifteen distinct co-expression modules (with the gene number greater than 200) were defined based on the obtained results (Figure 4A). Several co-expression modules comprised highly expressed genes at a given time point (Figure 4C). For example, the yellow module contained 1317 genes significantly enriched at 0.5 h after the salt treatment (Figure 4B). The brown and red modules comprised 2032 and 1054 genes highly expressed at 12 and 24 h after the salt treatment, respectively. In the yellow module, genes were mapped to the GO terms associated with cellular responses to stimulus, responses to organic substances and endogenous stimulus, cell communication, signaling, single organism signaling, and signal transduction. Moreover, KEGG enrichment analysis revealed an enrichment of the yellow module genes involved in plant-pathogen interactions, plant hormone signal transduction, MAPK signaling pathway, and phosphatidylinositol signaling system (Appendix A).

Specific terms relevant for stimuli responses, such as oxidoreductase activity and responses to stress, chemicals, abiotic stimulus, oxygen-containing compound, radiation, osmotic stress, oxidative stress, and reactive oxygen species, were abundant in the red module (Appendix A). This implies that the molecular responses mainly occurred after 24 h when the annual ryegrass was stressed by salinity. The GO terms related to cellular components, including membrane, localization, the establishment of localization, and transport, were also identified in the brown module. The metabolic pathway was the most significantly enriched KEGG term in the brown module (Appendix A). It revealed a programmed regulation in plants under salinity stress, demonstrating that the three modules may have independent regulatory networks and thus respond to different physiological processes.

We also identified several modules (cyan, pink, blue, purple, magenta, midnight blue, tan, and salmon) comprised of highly expressed genes in more than one sampling point (Figure 4C). The genes in cyan, pink, blue, and purple modules showed high expressions at 6 h and relatively lower expressions at 2 and 12 h after the salt treatment. The expression patterns of the genes in these modules exhibited similar profiles with the physiological indexes, indicating that 6 h might be a pivotal time node for a short-term salinity stress response. Meanwhile, the salmon and tan modules comprised highly expressed genes at 12 and 24 and had peak expression at 0.5 and 12 h post salinity stress. In addition, black and green modules showed unique expression profiles of the genes highly expressed at three-time points (2, 6, and 12 h) in the SS genotype and two-time points (6 and 12 h) in the SI genotype after salinity treatment. Gene expression peaks of the black and green modules occurred earlier in the SI genotype compared to the SS genotype. These two modules had many genes which were significantly enriched in the organelle subcompartment, plastid thylakoid, photosynthetic electron transport chain, the establishment of localization, transport, protein localization, and protein transport. This provided clues for the response difference observed between SS and SI genotype s. The turquoise module genes were highly expressed at 6, 12, and 24 h in both SS and SI genotypes, but expression levels were relatively higher in the SS genotype than in the SI genotype at 12 h after the treatment.

### 2.6. Identification of the Hub Genes in Different Stages

Hub genes are the critical function representatives of each module. The highest number of hub genes was in supercluster 4 of the brown and turquoise modules (4393 hub genes), while supercluster 1 of the yellow module had the least number of hub genes (740 hub genes) (Appendix A). Genes involved in the “environmental adaptation” and “Plant hormone signal transduction” were also significantly enriched, including circadian rhythm, MAPK signaling pathway, and plant-pathogen interactions pathway. Functional categories of the unique genes encoding calcium-dependent protein kinase (*CPKs*), calmodulin (*CPs*), and calcium-binding protein (*CMLs*) also presented in this module as hub genes, of which *PTI1* had the highest edge numbers (Figure 4D). However, the genes associated with these pathways differed in the yellow module. After 0.5 h of the treatment, twelve hub genes, including two disease resistance proteins encoding gens (*RPM1*) and a UDP-glycosyltransferase gene (*UGT79*), were identified as differentially expressed genes in SS and SI genotypes. Pathways relevant to genetic information processing, such as “transport and catabolism”, “folding, sorting, and degradation”, and “translation”, were identified as functionally enriched pathways in supercluster 2 after 2 and 6 h of the salinity stress, indicating the occurrence of the intrinsic molecular responses. Three metabolism-associated genes (*DGK1*, *MCCA,* and *ANR*) were differently expressed in SS and SI genotypes at these stages. Hub genes relevant to “folding, sorting, and degradation”, “transport and catabolism”, “translation”, and “plant hormone signal transduction” were detected in superclusters 2 and 3, indicating function overlap at 6 and 12 h post salinity treatment. Moreover, the functional categories of the unique genes associated with “replication and repair”, “lipid metabolism”, “carbohydrate metabolism”, “amino acid metabolism”, and “metabolism of cofactors and vitamins” revealed a possible physiological function transition. Photosystem-regulated genes from the cytochrome family, including cytochrome P450, cytochrome b-c1, and cytochrome c oxidase assembly protein encoding gene *COX15*, were also found in abundance in supercluster 2 and 3. This implied that energy-related pathways might be involved in short-term salinity responses.

Supercluster 4 comprised hub genes highly expressed at 12 h post salinity treatment. Many hub genes were significantly enriched in the “transcription” functional category, such as spliceosome processing, RNA polymerase activities, mRNA surveillance pathway, and RNA transport. The functional categories of unique genes related to the metabolism of cofactors and vitamins, carbohydrates, amino acids, and energy were also enriched in supercluster 4. The phytohormone signal transduction pathway contained many the hub genes, including ethylene-related hub genes (*ELI3*, *EIL1*, *ERS1*, and *EBF2*) and auxin-related hub genes (*ARF2*, *SAUR32*, *IAA17*, *IAA18*, and *ARF7*), which participated in the environmental information processing. Superclusters 4 and 5 had similar functional categories, which differed in the number of hub genes they contained. These results indicated that annual ryegrass rebalances after initial salinity stress. Many differentially expressed hub genes were identified from 6 h (20 DEGs) to 12 h (18 DEGs) after the salinity stress, indicating the influence of the stress preliminary stages on the salinity tolerance of SS and SI genotypes of annual ryegrass.

### 2.7. Identification of the Transcription Factors

We identified 1474 TF members of which, the majority were represented by the ERF (139) followed by FAR1 (137), C2H2 (104), and bHLH (98) (Figure 5A). According to the WGCNA results, 589 TFs belonging to 55 families were identified in 15 modules, and the most abundant TFs (100) were found in the yellow module (Figure 5C). About 75 TFs (27 TFs families) exhibited an ascending expression trend in the yellow module, whereas 25 TFs declined drastically at 0.5 h post salinity treatment. The TFs families, WRKY and NAC, have been shown to play crucial functions in plant abiotic stress responses [23]. Accordingly, over 35 members of WRKY and NAC, including WRKY6, WRKY11, WRKY17, WRKY 24, NAC022, NAC048, and the likes, were identified in the yellow module. The MYB/MYB-related genes, ERFs, and bHLHs have also been shown to protect plants from damage by activating hormone signaling during abiotic stress [24]. We identified various MYB-related genes, including *MYB1R1*, *MYB2*, *MYB61*, *EFR4*, *ERF7*, *ERF8*, *bHLH13*, and *bHLH148*, which were also highly expressed after 0.5 h of the salinity treatment. A total of 13 GATA members were identified in all modules, and nearly half of them were found in the yellow module, with the higher expression being at 0.5 h post salinity treatment in both SS and SI genotypes (Figure 5C). These stage-specific expression profiles and dynamics hint at the key functions of these TFs in the short-term salinity stress response in the annual ryegrass.

More TFs (37 TFs families) were detected with expression peaks at 2 and 6 h in the cyan module and 2, 6, and 12 h in the black and green modules (Figure 5C). The number of TFs in the different families were relatively scattered in the cyan, black, and green modules, with none being dominant over the other in number. Thus, the expression of various types of TFs during this period indicates the activation of complex defense mechanisms upon the translation of the environmental stimuli into internal signals. Various FAR1, bHLH, MYB/MYB-related, and C2H2 TFs, which modulate chlorophyll biosynthesis or hormone signaling, were also identified in cyan, black, and green modules [25,26]. Furthermore, pink, blue, and purple modules comprised the TFs similar to those in supercluster 3, with the higher expression being at 6 and 12 h post salinity treatment. The blue module consisted of 11 FAR1 family genes, including *FAR1*-*RELATED SEQUENCE 5*, *6,* and *9*, known phytochrome-interacting factors in chlorophyll biosynthesis [27]. The expression of three *AP2*, four *ARF*, and five *ERF* members increased at 6 and 12 h post salinity treatment, demonstrating potential functions of phytohormones in specific stages of the salt stress response. Various bHLH, MYB/MYB-related, and C2H2 family members were also identified in supercluster 3, but with relatively lower expression in the SI than the SS genotype, at 6 h post salinity treatment. Two zinc finger proteins, B-box zinc finger protein 25 (Unigene0089208) and zinc finger protein MAGPIE (Unigene0005150), were differentially expressed at 0.5 h in SS and SI genotypes [28]. In contrast to many signaling hormones and regulators, zinc-finger protein responds to stress non-specifically. Several important TFs, such as ERF 1 (Unigene0089208), MYB48 (Unigene0039808), and TGAL10 (Unigene0072722), were also identified as DEGs at 2 and 12 h post salinity stress.

The brown and turquoise modules showed a specific expression peak at 12 h post salinity treatment (Figure 5C). Moreover, genes in these modules contained TFs with similar functions in signal transduction, environmental information processing, and defense responses. The genes encoding SBP-domain transcription factors SPL2, SPL6, and SPL15, which may be relevant to several environmental stimuli, including salinity, were identified in the brown module [29]. Eleven genes encoding the transcription factors involved in the signaling of hormones, such as auxin and ethylene, were also present in the brown module and included *ARF5*, *ARF2*, *ERF1*, *ERF5*, and *ETHYLENE INSENSITIVE 3*. Various TFs encoding zinc finger CCCH/C2H2 domain-containing proteins were detected in the turquoise module. Previous studies have demonstrated that CCCH/C2H2 TFs affect ABA-, gibberellin- and sugar-mediated growth and stress responses [30,31]. Eight TFs, including trihelix transcription factor GT-2 (Unigene0006882), transcription factor ILI5 (Unigene0015540), and LOB domain-containing protein 40 (Unigene0029411), were differentially expressed in SS and SI genotypes at 0.5 h post salinity stress. Conversely, transcription factor bHLH87 (Unigene0060282) and B3 domain-containing protein (Unigene0103224) were differentially expressed at 2 h post salinity stress. The *FAR1-RELATED SEQUENCE 5* (Unigene0059999) was differentially expressed at 6 and 12 h post salinity stress. In contrast, squamosa promoter-binding-like protein 2 (Unigene0034224) and the NAC domain-containing protein 15 (Unigene0046406) were differentially expressed at 12 and 24 h, respectively, post salinity stress. These results show the differences in transcriptional regulation between the SS and SI genotypes at specific stages.

Supercluster 5 exhibited an expression peak at 24 h post salinity treatment in midnight blue, red, and tan modules (Figure 5C). Various MYB/MYB-related, bHLH, and NAC transcription factors were also overrepresented in these modules. A few FAR1-like transcription factors were enriched in these modules compared to clusters 2 and 3, indicating that the photosystem may be significantly involved in the early stages of salt stress. Most leucine zippers (bZIPs) and homeodomain leucine zipper (HD-ZIPs) TFs are relevant in general abiotic stress responses, induced by light signaling or salinity stresses, and they are post-transcriptionally activated by abscisic acid (ABA) signaling [32]. They were also strongly expressed at 24 h post salinity treatment. These results revealed that TFs might function at specific response stages in salinity stress. Some TFs showed different expression preferences between SS and SI genotypes, even if the expression peak occurred simultaneously.

### 2.8. Quantitative PCR (qPCR) Verification

Nine DEGs that were significantly up-regulated and down-regulated were selected for verification using qRT-PCR to determine the reliability of the transcriptome sequencing data. These DEGs included Unigene0016117 (TUB-transcription factor 14, Appendix A), Unigene0036579 (Ago, Appendix A), Unigene0102764 (Atg3, Appendix A), Unigene0032871 (Odc, Appendix A), Unigene0032817 (alkane hydroxylase MAH1-like, Appendix A), Unigene0062531 (Jasmonate-induced protein, Appendix A), Unigene0003785 (Prr, Appendix A), Unigene0028763 (Hsp4, Appendix A), and Unigene0095708 (WRKY1, Appendix A). The DEGs were found to be consistent between RNA-seq and qRT-PCR data, indicating the reliability of our transcriptome sequencing data.

## 3. Discussion

Climate change and improper irrigation lead to soil salinity, a threat to agricultural production globally. Soil salinization causes yield loss by affecting both the vegetative and reproductive growth of most crops. Therefore, crop acclimation to salinity stress through natural selection or artificial modification is important for mitigating this problem. Although annual ryegrass is tolerant to salinity stress, its physiological and molecular responses to salinity are well not understood. The present study determines the initial responses of annual ryegrass during the early stages of salinity stress. Two genotypes and six-time points were chosen for investigating the key candidates involved in temporal responses to salt stress and understanding salt tolerance mechanisms of the annual ryegrass.

### 3.1. Regulation of Ionic Balance

Enhanced uptake of Na^+^ and Cl^-^ under excessive salt conditions results in reduced water potential in plants. Concurrently, the uptake of cations, such as Ca^2+^ and K^+^ decreases, leading to nutrient imbalance and physiological or molecular function disorder in plants [33]. The production and accumulation of ROS have been proved to cause direct physiological function damage, which can be used as evident markers of oxidative stress-induced salinity stress [34]. Additionally, CAT, SOD, and POD levels were demonstrated to be significantly elevated under salt stress in ROS scavenging [35]. We analyzed several physiological indicators, including MDA, REL, and RWC, to assess the tolerance and trends in two genotypes at different stages of the salt stress. Almost all of these indicators showed similar profiles, which peaked promptly and increased continuously after a sharp decline following the salinity treatment. These results revealed the early stages of self-regulation or self-adaptation during salinity stress. Thus, the efficiency and response rates of this process may be important indicators for distinguishing the salinity-sensitive and salinity-insensitive genotypes of annual ryegrass. Another potential explanation for these physiological index curves could be the hormesis effect which occurs when plants are exposed to abiotic stress [36]. The basic levels of salinity stress induce adaptive responses in annual ryegrass, which intensify under severe salinity stress. This phenomenon is associated with dynamic regulations, especially in the ROS scavenging processes [37]. Moreover, the damage will not occur before a certain dose threshold is attained. This threshold determines the salt tolerance among different crop species or genotypes.

Ionic balance, especially for Na^+^ and K^+^ ions, is important for regulating plant responses to salt stress. For example, K^+^ ions play a vital role in activating many central enzymes, preserving cell turgor pressure, synthesis of protein and starch, metabolism of plant photosynthetic, and maintaining osmoregulation [38]. Sodium ions compete with K^+^ ions for the same transport channels of the plasma membranes, leading to inadequate water availability and ion toxicity. However, there are several strategies of Na^+^ reduction in plants, including Na^+^ uptake inhibition, Na^+^ efflux, and Na^+^ compartmentalization [39]. Therefore, the ability to maintain the cellular Na^+^/K^+^ homeostasis reflects the salt tolerance [4,40]. We found that the Na^+^/K^+^ ratio increased slowly before 6 h, then rose rapidly from 6 to 12 h, and relatively stabilized after 12 h following the stress treatment in the SI genotype. Contrarily, the Na^+^/K^+^ ratio of the SS genotype increased continuously at all sampling points, showing the discrepancies in the Na^+^/K^+^ homeostasis between the SS and SI genotypes. Various channels in plants mediate the Na^+^ efflux and K^+^ influx. These include sodium/hydrogen exchangers (*NHXs*), cation transporter (*HKT1*), cations/H^+^ antiporter (*CHX20*), potassium channel-related genes, potassium transporter (*HAKs*), vacuolar cation/proton exchangers (*CAXs*), chloride channel proteins (*CLCs*), and cyclic nucleotide-gated ion channel-related genes (*CNGCs*) [41]. The *HKT1* is a typical cation transporter that is selective for Na^+^, and studies have reported that the loss of function of *ZmHK1* increased the accumulation of Na^+^ in maize leaves [42,43]. It was also demonstrated that *HKT1* transcripts were significantly inhibited in salt-insensitive genotype of rice [44]. In our study, high expression of *HKT1* in the SS genotype may have led to Na^+^/K^+^ imbalance and the subsequent difference in salt resistance. Over 50 Na^+^/K^+^ homeostasis-related genes were identified in our study, but the differentially expressed homeostasis-related genes were only identified in the *HKT1* of the SS and SI genotypes. We speculate that *HKT1* may function in the early stages or be involved in salt stress sensing pathways dependent on or independent of salt stress. The high-affinity potassium ion transport protein, *HAKs*, is a putative Na^+^/H^+^ exchanger which transports Na^+^ from the cytoplasm to the vacuole driven by the H^+^ gradient [45]. In our study, most *HAKs* were upregulated under salt stress, consistent with previous reports [40].

Water deficits caused by high concentrations of Na^+^ directly or indirectly affect photosynthetic functions in plants [46]. We found that the DEGs were significantly enriched in the SS and SI genotypes in terms of the respiratory electron transport chain, electron carrier activity, and photosynthetic electron transport chain at 2 and 6 h post saline stress. This showed the photosystem damage associated with a high accumulation of Na^+^ accumulation. Moreover, the photosystem-related DEGs were overrepresented earlier (at 2 h) than the Na^+^/K^+^ imbalance, indicating that there may be other factors affecting photosystem stabilization. Several hub genes, including cytochrome P450, cytochrome b-c1, and cytochrome c oxidase assembly protein COX15 identified after 6 and 12 h following the salinity stress, were negatively influenced the photosystem externally [47].

### 3.2. Salinity Stress Sensing

While the physiological mechanisms of the ion-mediating systems have been well-described, less is known about the mechanisms by which plants perceive and respond to saline stress. Plants possess tissue- and species-specific sensing mechanisms operating at different timescales [48]. In our experiment, the sensing period of the saline stress signal was from 0.5 to 24 h). A previous study demonstrated that Ca^2+^, ROS accumulation, and hormone signaling are also part of the early signaling responses to stress stimuli [18]. Calcium (Ca^2+^) signaling plays a pivotal role in saline stress responses through the spatio-temporal changes of cytosolic Ca^2+^ concentrations. Moreover, calcium-dependent protein kinases (*CPKs*), including calmodulins (*CaMs*) and calcium-binding protein (*CBLs*), are key Ca^2+^ sensors involved in the stress sensing network, in which *CPKs* also function as Ca^2+^ responders [49]. In the yellow module, numerous *CPKs* (*CPK1*, *CPK4*, *CPK9*, *CPK10*, *CPK13*, *CPK15*, and *CPK20*) calmodulin-binding receptor (*CRCK2*), and calmodulin-binding transcription activator 3 (*CAMTA3*) were identified as hub genes. Their expression was upregulated at 0.5 h post saline stress, suggesting that Ca^2+^ is crucial for the multiple signal transduction [50,51]. It has been reported that over-accumulation of Na+ accumulating triggers cytosolic Ca^2+^ signaling during salt stress [52]. Accordingly, *Arabidopsis* roots were shown to perceive excess Na+ and induce Na+ efflux in soil within the 10 min of saline stress exposure [40]. These findings suggest that genes involved in Ca^2+^ signaling are activated earlier than 0.5 h (our first sampling point). The calcineurin B-like protein (*CBL*s) and CBL-interacting protein kinase (*CIPK*s) decode the Ca^2+^ signal elicited by salt stress. Consequently, *CIPK*-*CBL* complexes regulate SOS1 (salt overly sensitive) and establish an SOS signaling pathway [20]. Therefore, high expression of *CIPK19*, *CIPK21*, *CIPK26*, and *CIPK29* at 0.5 h after the saline stress indicates the co-occurrence of the diversiform pathways, which sense salt stress during the early stages.

Phytohormones are essential endogenous molecules that mediate plant saline stress through various physiological and biochemical mechanisms [53]. As a second messenger, calcium-coupled *CIPK*-*CBL* complexes activate phytohormone signaling in response to saline stress [54]. Additionally, the complex acts as an indispensable signaling compound for integrating and coordinating intricate hormone pathways, thus facilitating stress signal recognition by plants [55]. Ethylene is an important stress hormone conferring salinity tolerance in plants by preventing ROS accumulation [56]. The effects of Ca^2+^ on the biosynthesis of endogenous ethylene could promote plant root morphology changes converting under salt stress [57]. However, a previous study suggested that ethylene is not the dominant component of the hormone-mediated responses to salinity stress [58]. Nevertheless, several *ERF*s have been demonstrated to regulate the downstream signaling networks involved in the adaption to oxygen deprivation within a timescale of minutes to hours [59,60]. In this study, the most abundant *ERF*s and *EIL*s were positively regulated by ethylene signaling and highly expressed at 0.5 h post saline stress. This suggests that ethylene may function as a receptor for the early signaling events in salt stress response [61]. Abscisic acid is the central mediator regulating stomatal closure, ion homeostasis, and stress-related genes transcription during salt stress [53]. However, an increase in ABA accumulation has been demonstrated to reduce ethylene production, thus showing the counteractions of ABA on ethylene during salt stress [55]. Additionally, ABA was shown to enhance water uptake and water potential via numerous ABA-dependent TFs, such as bZIPs and bHLH families [62,63], which significantly enriched at 24 h post saline stress. This finding indicates that ABA and ethylene function at different stages in salt response. Studies have reported that SOS signaling induces auxin redistribution in roots, thereby mitigating salinity stress [64]; however, auxin concentrations are antagonistically regulated by ABA [65]. Our results show that various auxin-responsive proteins (IAAs) and auxin response factors (ARFs) were highly expressed in superclusters 2 and s 4 at 2, 6 and 12 h post saline treatment. This suggests that salinity response involves ethylene, ABA, and auxins at different time points. Several members of the NAC family have also been shown to regulate salt signaling cascade via auxin-related genes [66]. Besides the mentioned phytohormones, we also identified jasmonic acid (JA) and salicylic acid (SA) genes and associated pathways in almost all early stages of salt response. These two phytohormones (JA and SA) promote salt stress tolerance by enhancing the antioxidant system, inhibiting root elongation, maintaining osmotic pressure, and promoting photosynthesis [67,68,69]. Cytokinins (CKs) and brassinosteroids (BRs) also important in salt response [70]; however, we could not demonstrate their involvement in the early stages of salt signaling. In general, salinity perturbs plant hormone balance; however, the crosstalk between various phytohormones mediates signaling cascade against abiotic stresses, such as salt stress, in a short period upon exposure.

### 3.3. Transcriptional Regulation

Plants adjust to environmental fluctuations by constantly altering gene expression profiles throughout their life cycle. Transcription factors construct the multi-hierarchical networks with specific temporal dynamics of salt stress tolerance. We identified the WRKY, NAC, and ERF TFs during the early stages of the salt treatment. Various members of the WRKY family (*GhWRKY41*, *GhWRKY34*, *DgWRKY5*, *SlWRKY3*, *DnWRKY11*, and *ZmWRKY86*) have been shown to enhance salt stress tolerance in plants like cotton [71,72], chrysanthemum [73], tomato [74], tobacco [75], and maize [76]. Additionally, other WRKY members, including *WRKY18*, *WRKY40*, and *WRKY60*, negatively affect ABA signaling by repressing the ABA-responsive genes (*ABI4* and *ABI5*) [77]. Conversely, *GhWRKY6*, a member of the WRKY family, was reported to enhance cotton salt tolerance by activating ABA signaling and scavenging for reactive oxygen species, indicating that WRKY genes have diverse functions in plant salt responses [78]. A recent study has revealed that the feedback loops formed by the WRKY members, and their targets may serve as dominant nodes in the regulatory networks of salt-responsive genes [23]. The responses are divided into two distinct groups, the early- and late-response groups, defined by their time-dependent functions. The early-response group related to the ABA signaling pathway was shown to be highly linked to *WRKY3*, *WRKY6*, *WRKY7*, and *WRKY9* [23]. This result is consistent with our finding which indicated that various WRKY TFs were identified at 0.5 h after subjecting the annual ryegrass to salt stress. Furthermore, a portion of NAC members, including *NAC5*, *NAC6,* and *NAC57*, have been reported to enhance tolerance to salt stress by maintaining osmotic balance and increasing ROS scavenging [79,80,81]. Other members, such as *NAC2* and *NAC022*, are involved in phytohormones signaling. For example, *OsNAC2* decreases auxin accumulation while increasing CK accumulation to regulate salt concentration. It was also shown that *OsNAC2* positively affects ABA-mediated salt tolerance [82]. Among TFs members, *ERFs* were the most abundantly expressed. Previous studies demonstrated that the *ERFs* play crucial roles in saline stress and ethylene responses by specifically binding the GCC-box to the downstream promoters [83]. Evidence has revealed this is activated by the ethylene biosynthesis and accumulation activates the downstream signaling network, thus altering the expression of *ERFs* [84]. The seven *CPKs* members exhibited high expression patterns similar to *ERFs* at 0.5 h of exposure to saline conditions. This finding enabled us to identify the salt-responsive genes and determine the ethylene-responsive genes during salt stress.

Phytochrome A is one of the primary photoreceptors responsible for the far-red light response [85] regulated by *FAR-RED ELONGATED HYPOCOTYL3* (*FHY3*) and its homolog *FAR-RED-IMPAIRED RESPONSE1* (*FAR1*). The *FAR1* functions in plant photosystem developments, such as chloroplast division, UV-B light response, and chlorophyll biosynthesis regulation. Additionally, *FAR1* suppresses oxidative stress by directly targeting *MIPS1* and *MIPS2* during salt stress [86]. Lipid metabolism and starch synthesis have also been associated with *FAR1* [87,88]. We identified 27 differentially expressed *FAR1* TFs from 2 h to 6 h after saline exposure. Meanwhile, the SOD, CAT, and POD activities, and MDA concentrations declined in this phase, implying that the photosystem may be negatively affected by the excessive ROS. The *FAR1* TFs relieve the negative effects by balancing growth and environmental stress in plants. Furthermore, bHLH, another important member of the TF family, has been reported to regulate ABA signaling [89]. These bHLHs can act as core nodes that integrate gibberellin, jasmonate, brassinosteroid, and auxin pathways in the plant hormones signaling network [89]. These results uncovered crucial regulators which ensure the stability of the photosystem and efficiency of the ROS scavenging processes. The activities of the SOD, CAT, and POD and concentrations of MDA increased between 12 and 24 h following the physiological disinhibition. The SPL, ARF, and C3H/C2H2 showed higher expressions after 12 h, while MYB, bZIP, and HD-ZIP TF families exhibited higher expressions after 24 h.

Some SPL-encoding proteins have been implicated in salt stress responses by interacting with salt-inducible miRNAs [90]. Lower concentrations of NaCl induced the inhibition of *TAS3*, an *ARF3*/*4* repressor, by miR390, but enhanced salt tolerance by promoting the expression of *ARF3*/*4*. Contrarily, high concentrations of NaCl reduced salt tolerance due to the reversed regulation pathway [91]. The MYB TFs are prominent regulators of salt stress tolerance levels. Studies have reported that MYB TFs regulate salt stress tolerance by enhancing ROS scavenging ability [92], regulating flavonoid biosynthesis [93], regulating the epigenetic modification [94], and integrating the ABA signaling [95]. These results show that various members of the TF family function in salt stress responses via different mechanisms. Moreover, the differentially expressed TFs aided the identification of the key mediators of saline stress regulation. However, further studies are needed to assess the TFs implicated in the early phases (such as WRKY, NAC, and ERF), mid-stages (FAR1, bHLH, ARF, and ERF), and late stages (MYB, SPL, C3H/C2H2, bZIP, and HD-ZIP) of salt stress responses. The WRKYs, ERFs, and MYBs emerged as key regulators of salt tolerance in annual ryegrass.

### 3.4. Metabolism Regulation

Hub-gene identification revealed that metabolism-related lipids and carbohydrates were involved in salt stress response after 6 h of the salinity treatment. Salt stress limited the photosynthetic ability of the plants and induced oxidative damage due to the excessive accumulation of ROS, thus causing an imbalance between carbohydrate synthesis and decomposition [96]. The stress also influences the translocation of assimilates by inducing over-accumulation of carbohydrates in the leaves, thus resulting in photosynthetic feedback repression [97]. Contrarily, the carbohydrates transported from the source to sink tissues provide more storage energy in the roots, enhancing salt stress tolerance by strengthening osmotic adjustment and ROS scavenging. Previous studies have demonstrated that salinity stress significantly alleviates the PSII electron transport and photosynthetic oxygen evolution, causing rapid changes in energy metabolism [98,99]. Lipids are the dominant constituents of the membrane system, and stable membrane transport is important for photosynthesis and intercellular communications [100]. Lipid biosynthesis, transport, accumulation, and degradation processes are important in regulating the integrity, permeability, and fluidity of membrane structure, which modulates salinity stress tolerance in plants [101,102]. Lipid peroxidation and membrane disruption are the main damages caused by ROS accumulation [103]. Vitamins are antioxidants that directly scavenge for the ROS, thereby quenching the chain reactions of lipid peroxidation before oxidative damage occurs [104]. Plants maintain the energy supply by repressing aboveground growth following saline signaling transduction and photosynthesis inhibition. Various metabolism processes may also be involved in the late phases of short-term responses to salt stress.

## 4. Materials and Methods

### 4.1. Determination of Relative Water Content

Leaf samples (approximately 0.1 g) were collected at 0, 0.5, 2, 6, 12, 24, 48, and 72 h after salt treatment. Six biological replicates of each sample were collected for relative water content determination. Thereafter, the samples were wrapped with absorbent paper and soaked for 24 h in 50 mL centrifuge tubes filled with water. The leaves were then dried and weighed for the saturated fresh weight. Subsequently, dry weight was measured after baking the leaves at 75 °C to a constant weight. The measuring was repeated 4 times, and the values were averaged. The relative water content was then calculated as follows: Relative water content (%) = (fresh weight − dry weight)/(saturated fresh weight − dry weight) × 100% as described previously [105].

### 4.2. Determination of Electrolyte Leakage

Electrolyte leakage was measured using the conventional conductivity meter method [106]. Six biological replicates of each sample were collected for electrolyte leakage determination. Briefly, 0.1 g of the leaves were cut into pieces and soaked for 24 h at room temperature in 10 mL tubes containing deionized water. A conductivity meter was used to measure the electrical conductivity before (S1) and after (S2) heating the samples in a boiling water bath for 15 min (the samples were cooled to room temperature before taking the measurement). Thereafter, the electrolyte leakage was calculated as follows: Relative conductivity = (pre-heating conductivity S1/post-heating conductivity S2) × 100%.

### 4.3. Determination of Malondialdehyde Content and the Protective Antioxidant Enzyme Activity

Approximately 0.1 g of fresh leaves were collected, freeze-dried in liquid nitrogen and ground. The ground samples were transferred into centrifuge tubes, and 1mL of the pre-cooled phosphoric acid buffer was added. After that, the samples were centrifuged at a speed of 15,000× *g*/rpm for 20 min at 4 °C, and the supernatant was collected as crude enzyme extract. For the MDA content determination, 1 mL of the reaction solution was added to 0.5 mL of crude enzyme solution and heated on a water bath for 30 min at 95 °C. The mixture was then cooled to room temperature and centrifuged for 10 min at 10,000× *g*/rpm. The absorbance value of the obtained supernatant was measured at 532 nm and 600 nm. The MDA value was obtained by subtracting the absorbance values of the crude enzyme extract from that of the reaction mix supernatant [107]. The content of MDA was expressed as nmol per g fresh leaf (nmol gFW ^–1^).

The activity of SOD was measured using the riboflavin-NBT method [107]. Briefly, 0.5 mL of crude enzyme extract was added to 500 mM potassium phosphate buffer (pH = 7.8) containing 20 µM riboflavin, 750 µM NBT, 130 mM methionine, and 1000 µM EDTA. The absorbance of NBT photoreduction (formation of blue formazan) was then measured at 560 nm. One unit of SOD activity was defined as the amount of crude enzyme extract required to inhibit 50% of the NBT photoreduction compared with the control reaction. The results were expressed as units of SOD activity per mg fresh leaf per min (U mg^−1^ min^−1^). Meanwhile, the POD activity was measured by adding 0.1 mL of crude enzyme extract 8 mL solution containing 80 mM guaiacol, 80 mM H_2_O_2_, and 200 mM potassium phosphate buffer (pH = 6.8). After the reaction was stopped by adding 2 mL of 5% (*v/v*) H2SO4, the absorbance value was read at 480 nm and expressed as a unit of POD activity per mg fresh leaf per min (U mg^−1^ min^−1^) [108]. Catalase (CAT) activity was measured by adding 100 µL crude enzyme extract to 6 mL of a solution consisting of 100 mM potassium phosphate buffer (pH = 7.0) and 40 mM H_2_O_2_. The result was indicated by a reduced absorbance value at 240 nm and expressed as unit of CAT activity per mg fresh leaf per min (U mg^−1^ min^−1^) [108].

Six biological replicates of each sample were collected for MDA, SOD, POD, and CAT determination. The physiological data were assessed via Duncan’s multiple range tests in SPSS software (version 13.0), and significant differences between SS and SI genotypes were shown by *p* < 0.05 and *p* < 0.01.

### 4.4. Determination of the Na^+^ and K^+^ Concentrations

Three biological replicates of each sample were collected and dried. Dried samples (0.4 g accurately measured to four decimal places) were weighed into the inner vial of the Teflon digestion tank and soaked overnight in 5 mL of nitric acid. The inner vial was covered thereafter, and the stainless-steel coat tightened. A thermostatic drying oven was then used to dry the samples at 80 and 120 °C for 2 h, and at 160 °C for 4 h. The samples were cooled to room temperature, opened under a fume chamber, and heated until the acid evaporated. The obtained digestive liquid was transferred into a 25 mL volumetric flask, and a small amount of nitric acid solution (1%) was used to rinse the residual samples (3 times) from vessels (the inner jar and its cover) into the volumetric flask. The sample was then scaled up using 1% nitric acid forming the test liquid. A blank reagent test was conducted before analyzing the test liquid using an inductively coupled plasma- emission spectrometer (Thermo Fischer Scientific, Waltham, MA, USA) [109].

### 4.5. RNA Extraction, Library Construction, and Sequencing

Samples were selected from the SS and SI genotypes at 0, 0.5, 2, 6, 12, and 24 h after treatment for RNA-seq. Three biological replicates of each sample were collected and immediately stored in liquid nitrogen before use. Total RNA was extracted using the Trizol reagent kit (Invitrogen, Carlsbad, CA, USA) and treated with RNAse-free deoxyribonuclease (DNAse) (Qiagen, Valencia, CA, USA). Agilent 2100 Bioanalyzer (Agilent Technologies, Palo Alto, CA, USA) was used to determine the integrity and quality of the extracted RNA. After the extraction, eukaryotic mRNA was enriched using oligonucleotide beads, while the prokaryotic ribosomal RNA (rRNA) was removed using the ribo-zerotm magnetic kit (Epicentre, Madison, WI, USA). The enriched mRNA was then fragmented using a lysis buffer and transcribed into complementary DNA (cDNA) using a random primer. We constructed 36 cDNA libraries. Thereafter, a QiaQuick PCR extraction kit (Qiagen, Venlo, Netherlands) was used to purify the DNA fragments, which were then end-repaired, polyadenylated, and ligated to Illumina sequencing adapters. The ligated products were then separated by agarose gel electrophoresis and amplified using PCR before sequencing using Illumina HiSeqTM 4000 at Gene Denovo Biotech (Guangzhou, China).

### 4.6. Identification, de Novo Assembly, and Determination of the Expression Levels of the DEGs

Clean raw data were obtained by removing the reads containing (i) adapters, (ii) unknown “N” bases, and (iii) low-quality bases. Subsequently, gas chromatography and calculation of the Q20 and Q30 of the reads were conducted. De novo transcriptome assembly was carried out for the reference genome using Trinity software [110]. An FPKM (fragment per kilobase of transcript per million mapped reads) value was calculated using StringTie software to quantify the expression abundance and variations of the unigenes [111,112]. Meanwhile, DESeq2 and EdgeR software were used to analyze the differential expression of RNAs between two different groups [113,114]. Genes with a false discovery rate (FDR) lower than 0.05 and absolute folding change greater than or equal to 2 were considered differentially expressed genes.

The weighted gene co-expression network analysis was conducted using the WGCNA package (v 3.3.0) as described by Langfelder [115]. A total of 17,180 genes with FPKM values greater than 2 were chosen in more than half the samples for the analysis. The modules were obtained via blockwiseModules with the soft thresholding power of 8 and MinModuleSize of 50. The networks were visualized using Cytoscape (v 3.1.2) [116]. Hub genes identification was performed as described by Bahman Panahi [117].

### 4.7. Principal Component and Transcription Factors Analyses

A single protein-coding sequence was used as a query on the BLASTp search tool against the plant transcription factor database (TFdb) (http://planttfdb.gao-lab.org/, accessed on 14 June 2020) to predict the TF family. Principal component analysis (PCA) was conducted using the R package through omicsmart platform (www.omicsmart.com, accessed on 2 March 2020) to reveal the structure/relationship among the samples/data.

### 4.8. Gene Ontology and KEGG Enrichment Analyses of the DEGs

Gene ontology database (http://www.geneontology.org/, accessed on 5 March 2020) was used for the GO enrichment analysis to map all the DEGs according to the GO terms. Each GO term represented the number of genes determined through the hypergeometric test and whose genomic background differed significantly from the DEG-enriched GO terms [118]. Conversely, KEGG enrichment analysis was performed using the KEGG orthology-based annotation system (KOBAS) [119]. The calculated *p*-values for the GO terms and KEGG categories were corrected for FDR, and a threshold value of FDR ≤ 0.05 represented an important GO classification and KEGG pathway of the DEGs.

### 4.9. Real-Time Quantitative PCR (qRT-PCR) Validation

Nine DEGs were selected for verifying the reliability of the transcriptome data. The DEGs-specific primers were designed using Primer 5.0 software, and glyceraldehyde-3-phosphate dehydrogenase (GAPDH) was used as a reference gene. Each reaction volume was 20 μL containing 10 μL of SYBR green master mix, 0.4 μL of the forward and reverse primers each, and 2 μL of the cDNA. Amplification conditions were as follows: initial denaturation at 95 °C for 30 s, followed by 35 cycles of denaturation at 95 °C for 10 s, and annealing at 60 °C for 30 s. Subsequently, the delta–delta (2^−ΔΔCt^) method was used to calculate the relative expression of each sample [120]. The primer sequences are listed in Appendix A.

## 5. Conclusions

The RNA-seq data were obtained to investigate the profiles of the salinity response phases of the SS and SI genotypes of annual ryegrass. Stage-specific profiles revealed novel regulation mechanisms in salinity stress sensing, phytohormones signaling transduction, and transcriptional regulation of the early salinity responses. Moreover, we identified distinctively co-expressed gene clusters and highly ordered gene networks in different phases of the short-term salt stress response, which provide insights in identifying the key regulators of salt stress response in the annual ryegrass. These findings enhance our understanding of salt stress responses in the annual ryegrass and accelerate the breeding of salt-tolerant forages.

## Figures and Tables

**Figure 1 ijms-23-03279-f001:**
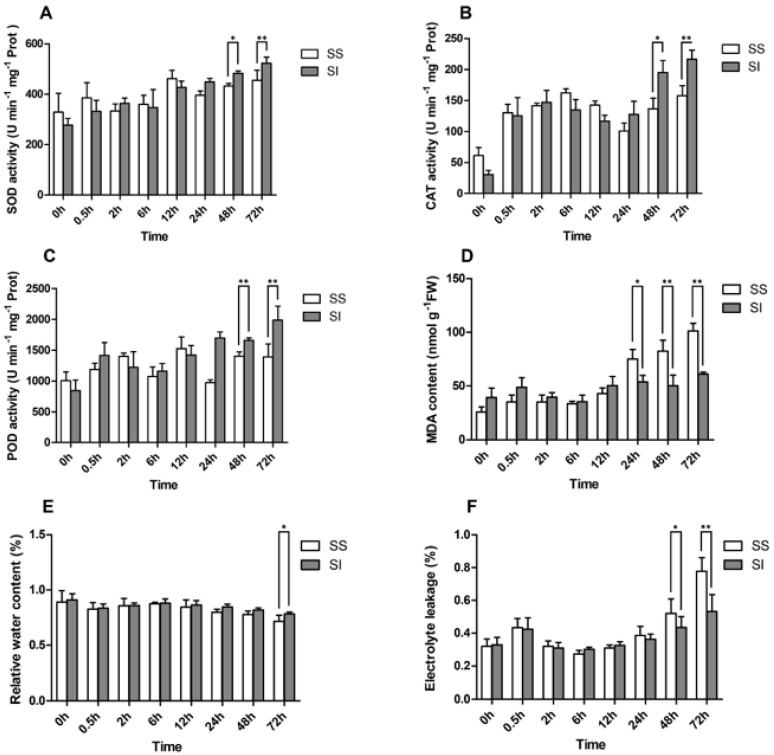
Activities of SOD (**A**), CAT (**B**), and (**C**) POD, MDA concentration (**D**), relative water content (**E**), and electrolyte leakage (**F**) in annual ryegrass leaves after salt treatments. The white columns represent salinity-sensitive (SS) genotypes, while the grey columns depict salinity-insensitive (SI) genotypes of the annual ryegrass. The ordinate axis indicated the mean values while the abscissa axis shows sampling time points (0.5, 2, 6, 12, 24, 48, and 72 h) after salt treatments. The symbols * and ** indicates significant difference at *p* < 0.05 and *p* < 0.01, respectively.

**Figure 2 ijms-23-03279-f002:**
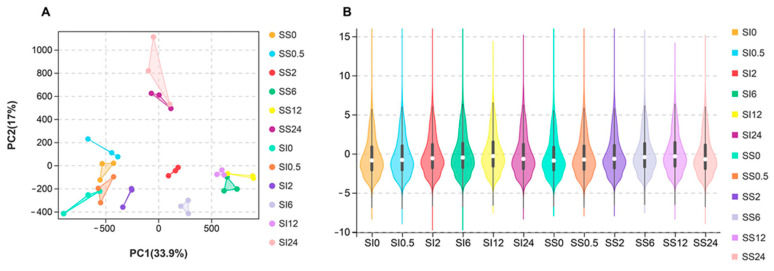
Transcriptional association and FPKM distribution between samples. (**A**) Principal component analysis of genes expressed across all the samples. (**B**) FPKM distribution of the genes expressed in various samples. SS represents salinity-sensitive genotypes, while SI denotes salinity-insensitive genotypes of the annual ryegrass.

**Figure 3 ijms-23-03279-f003:**
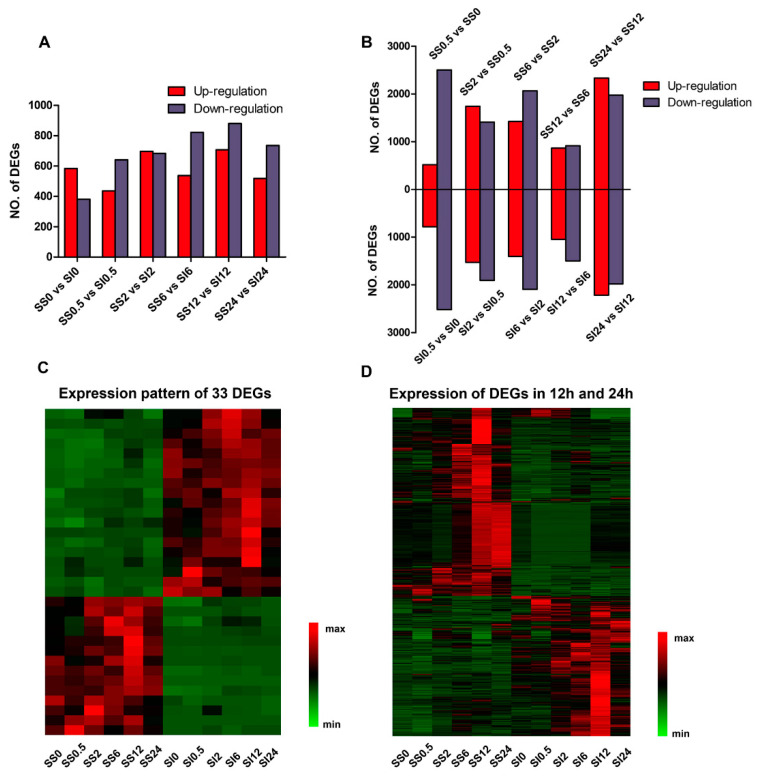
A summary of the differentially expressed genes (DEGs). (**A**) DEGs obtained from five comparisons. (**B**) DEGs from the adjacent sampling points. The abscissa axis represents the different comparisons, whereas the ordinate axis represents the gene number. (**C**) Expression pattern of 33 DEGs and (**D**) expression pattern of DEGs in 12 h and 24 h. Each row in the heat map indicates a gene. The red color denotes the high expression level, while the green color represents the low expression level. SS represents salinity-sensitive genotypes, whereas SI signifies salinity-insensitive (SI) genotypes of the annual ryegrass.

**Figure 4 ijms-23-03279-f004:**
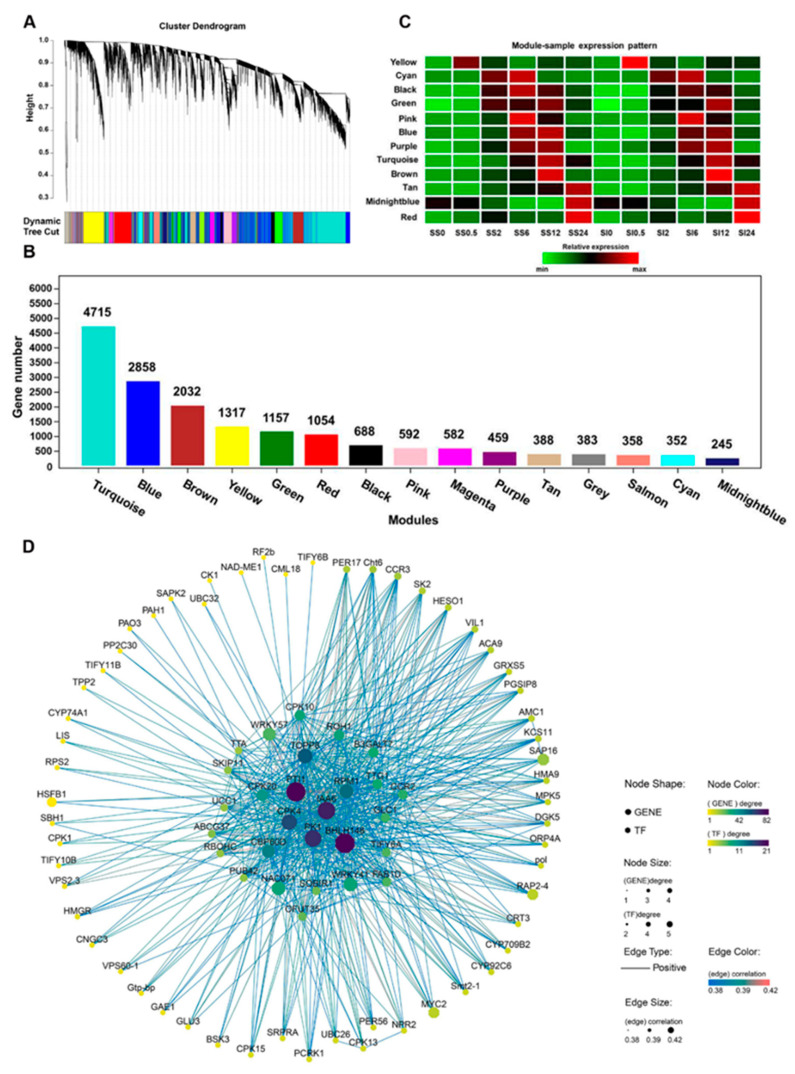
A weighted correlation network analysis of genes from 12 groups. (**A**) A hierarchical cluster tree showing the co-expression modules, identified via WGCNA. Each leaf in the tree represents one gene. Major tree branches constitute 15 modules labeled by different colors. (**B**) A histogram showing the gene numbers in different modules. (**C**) A heat map indicating the relative expression of the modules in six sampling points. The ordinate axis represents the different modules, while the abscissa axis denotes samples collected at 0.5, 2, 6, 12 and 24 h post salt treatments. Each row in the heat map indicates a gene. The red color denotes the high expression level, while green color represents the low expression level. (**D**) Co-expression subnetwork of the yellow module. The octagon node indicates transcription factor, whereas circular node represents genes. Node color and size indicates gene degree, while edge size and color indicate gene correlation. SS represents salinity-sensitive genotypes while SI represents salinity-insensitive (SI) genotypes of the annual ryegrass.

**Figure 5 ijms-23-03279-f005:**
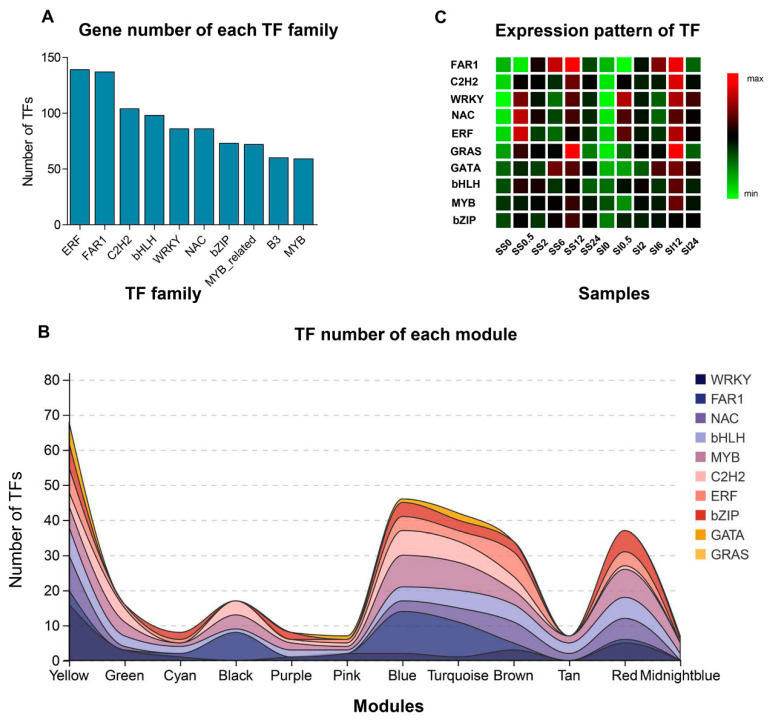
TFs identification during the early salt response of annual ryegrass. (**A**) Number of TFs in different families. The abscissa axis represents the different TF families, while the ordinate depicts TF numbers. (**B**) Number of TFs in different modules. The abscissa axis represents the different modules, whereas the ordinate shows TF numbers. (**C**) The expression pattern of TF families in the samples. The ordinate axis represents the different modules, while the abscissa denotes the samples collected at 0.5, 2, 6, 12, and 24 h post salt treatments. Each row in the heat map indicates the genes. The red color denotes the high expression level, while the green color represents the low expression level. SS represents salinity-sensitive while SI represents salinity-insensitive genotypes of the annual ryegrass.

## Data Availability

All the RNA-seq data of this study have been deposited into the NCBI with accession numbers SRR16085492, SRR16085494, SRR16085487, SRR16085488, SRR16085493, SRR16085489, SRR16085497, SRR16085496, SRR16085491, SRR16085490, SRR16085498, and SRR16085495. The plant materials were provided by the Department of Forage Science, College of Grassland Science and Technology, Sichuan Agricultural University, Chengdu, China.

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
