# Peer review of "Comprehensive Transcriptome Analysis Uncovers Distinct Expression Patterns Associated with Early Salinity Stress in Annual Ryegrass (Lolium Multiflorum L.)"

_ijms, 2022, doi:10.3390/ijms23063279_

Round 1

Reviewer 1 Report

It is ok now.

Reviewer 2 Report

The authors revised the paper as per my comments and improved this paper.

This manuscript is a resubmission of an earlier submission. The following is a list of the peer review reports and author responses from that submission.

Round 1

Reviewer 1 Report

Dear Authors,

The manuscript entitled Comprehensive transcriptome analysis uncovers distinct expression patterns associated with early salinity stress in Annual ryegrass (Lolium multiflorum L.) concerns a very important field of research. Salinity stress resulting from climate change and irresponsible human activities in the field of fertilization and land drainage affects many plant species including fodder plants. In general, the manuscript is written quite well, but it is difficult to assess its merits because the material and methods section lacks a description of the studied material and the way of conducting the experiment. There is only a detailed description of the analytical methods. Although it is not known how many replicates the analyses were performed in. Statistical analysis of phenotyping results is also missing.  Did the obtained results differ significantly depending on the duration of salinity stress? The description of RNAseq analysis lacks details of NGS conditions i.e. read length, number of reads per sample, biological repeats. There is also some inaccuracy in the text of the manuscript. The methodology states that the transcriptome was assembled de novo, while the description of the results shows that the reads were mapped to a reference genome. So my question is how was the analysis of the RNAseq results performed? What reference genome was used?
The description of the WGCNA results indicates that microarrays were used, the description of the methodology does not indicate the use of this technique. 
Figure 4 should be larger, it is not very readable. 
The descriptions for the figures should be self-explanatory.
On lines 648-654 the text is in italics.
As I have no way to evaluate the correctness of the experiments and the results obtained due to shortcomings in the manuscript, I think that the manuscript should be rejected in this version.

Best regards,

M.

Reviewer 2 Report

In this paper, the authors showed the comprehensive transcriptome analysis uncovers distinct expression patterns associated with early salinity stress.

The authors have taken many data and the presentation quality is good.

However, The authors should explain why only early salinity stress is considered. In the case of L.multiflorum later stage is also sensitive to stress. Only 72 h of stress is not representitive.

The paper is too wordy, especially the Introduction. Please concise it.

Abstract is also unnecessarily long.

The bar graph should contain statistical lettering or any indicators so that reader can see the comparison clearly.

Some of the references are old. Please update them.

Please explain the antioxidant defense in the light of recent and vital references. e.g. 

Int. J. Mol. Sci. 2021, 22(17), 9326; https://doi.org/10.3390/ijms22179326